# People with severe mental illness have low rates of screening for non-communicable diseases: Findings of a multi-country cross-sectional study in South Asia

## Research Article

comorbidity; South Asia; screening; non-communicable diseases; severe mental illness

**Corresponding author:**
Koralagamage Kavindu Appuhamy;
Email: kka505@york.ac.uk

Koralagamage Kavindu Appuhamy[1] (ID), Fraser Wiggins[1], Alex Mitchell[1], Helal Uddin Ahmed[2] (ID), Mark Ashworth[3], Faiza Aslam[4], Jan Boehnke[5], Olga Garcia[6], Richard I.G. Holt[7,8], Rumana Haque[9], Krishna Prasad Muliyala[10], Pratima Murthy[11], Asad Tamizuddin Nizami[4], Benjamin Perry[12], David Shiers[13], Najma Siddiqi[1], Kamran Siddiqi[1], Salim Virani[14] and Gerardo A. Zavala[1]

[1]Health Sciences, University of York, UK; [2]Faridpur Medical College, Bangladesh; [3]Life Course and Population Sciences, King's College London, UK; [4]Institute of Psychiatry, Rawalpindi Medical University, Pakistan; [5]Health Sciences, University of Dundee, UK; [6]Facultad de Ciencias Naturales, Universidad Autonoma de Queretaro, Mexico; [7]Human Development and Health, University of Southampton Faculty of Medicine, UK; [8]Southampton National Institute for Health Research Biomedical Research Centre, University Hospital Southampton NHS Foundation Trust, UK; [9]ARK Foundation, Bangladesh; [10]Psychiatry, National Institute of Mental Health and Neurosciences, India; [11]National Institute of Mental Health and Neurosciences, India; [12]School of Psychology, University of Birmingham, UK; [13]Psychosis Research Unit, Greater Manchester Mental Health NHS Foundation Trust, UK and [14]The Aga Khan University, Pakistan

## Abstract

People with severe mental illness (SMI) die 10–20 years earlier than the general population, largely due to non-communicable diseases (NCDs) such as hypertension and diabetes and risk factors such as hypercholesterolaemia. This cross-sectional study gathered data from people with SMI from three national mental health institutions in South Asia. Data was collected based on the WHO Stepwise approach to NCD risk factor surveillance and the prevalence of screening, diagnosis and treatment for diabetes, hypertension, and hypercholesterolaemia was assessed. Logistic regression models assessed the associations of sociodemographic characteristics with NCD screening. Three thousand nine hundred and eighty nine participants were recruited. Screening prevalence varied by country and disease, with hypertension being the most commonly screened NCD (Bangladesh = 52.5% [50.0–55.1], India = 43.1% [40.3–45.9], Pakistan = 60.9% [58.2–63.5]), and cholesterol was the least common (Bangladesh = 4.1% [3.2–5.2], India = 14.8% [12.9–17.0], Pakistan = 9.6% [8.1–11.3]). Characteristics such as BMI, age and education level were positively associated with screening, and females were more likely to be screened than males. There are low levels of screening for NCDs among individuals with SMI accessing tertiary institutions in South Asia, with significant sociodemographic disparities. Standardised screening protocols tailored to South Asian populations could mitigate the increased risk of NCDs in this population.

## Impact Statements

This study provides crucial information about the prevalence of screening for non-communicable diseases (NCDs) in people with severe mental illness (SMI) in South Asia, as this is a key area for further development to reduce the huge burden of physical health comorbidities on the higher mortality seen in people with SMI compared to the general population. As well as strengthening the evidence base for improving screening of NCDs in people with SMI, several key associations with screening uptake have been identified, which highlight some of the different characteristics that are more strongly or weakly associated with screening for NCDs. At a national level, this information may impact health policy as it can be utilised to help inform and guide the development of screening protocols that are tailored to the South Asian population. This will have the potential to identify NCDs such as diabetes at an earlier stage, when they can be more effectively managed and likely to be cost-saving to health services. Furthermore, by potentially shaping future policy, this research may contribute to improving health outcomes and consequently reducing the health inequalities experienced by the SMI population in South Asia.

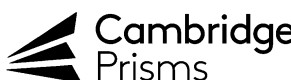

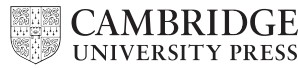

## Introduction

Severe mental illness (SMI) is a term used to define a group of psychiatric conditions which have particularly persistent symptoms and cause significant functional impact (Johnson, 1997). This umbrella term typically includes diagnoses such as schizophrenia spectrum disorders, bipolar disorder, and severe depression with psychotic symptoms (Zavala et al., 2020).

People with SMI die on average 10–20 years earlier than the general population (de Hert et al., 2009; Afzal et al., 2021). Non-communicable diseases (NCDs) are the biggest contributors to this excess mortality (de Hert et al., 2009; Afzal et al., 2021). People with SMI are particularly vulnerable to NCDs due to a combination of factors, including individual risk factors such as poor diet and physical inactivity, the high prevalence of obesity seen in this population (Zavala et al., 2023), genetic and environmental factors, healthcare inequalities, as well as adverse effects from antipsychotic treatment (Appuhamy et al., 2023), and mental health stigma (Vaishnav et al., 2023).

Research in the field of multi-morbidity in SMI has predominantly focused on Western populations. Yet, a growing body of research on non-Western populations, for example, people with SMI from South Asia, has clearly shown the additional barriers to healthcare access, and a lack of resources and awareness (Singh et al., 2019), which may widen the morbidity and mortality gap between individuals with SMI and the rest of the population. This disparity is likely to be more pronounced in individuals with SMI in the Indian subcontinent, as South Asians have a higher predisposition to obesity-related NCDs (OR-NCDs) compared to White European populations. This increased risk is attributed to factors such as higher abdominal adiposity and greater insulin resistance (Misra and Khurana, 2010; De Foo et al., 2022).

There is a growing global need to tackle the challenge of NCDs. For example, NCD screening has been noted as one of the targets in the United Nations Sustainable Development Goals. Screening for NCDs is cost-effective for the prevention of NCDs in low- and middle-income countries (LMICs), especially for high-risk groups (Sharma et al., 2022). Therefore, screening of NCDs is important to identify those most at risk in a timely manner for effective intervention to take place. In particular, the importance of managing NCDs in people with SMI is underpinned by the utilisation of screening tools in multiple higher-income countries (HICs). For example, the Lester Tool is now widely used in the National Health Service in the UK (NHS England, 2014), and the Health Improvement Profile is used in the USA, Hong Kong and Finland (Bos et al., 2022).

To appropriately address the physical health needs of people with SMI, screening for NCDs is important and may help to guide more targeted interventions. Also, determining which socio-demographic factors are associated with being screened for NCDs may help to guide more targeted screening programmes in the future to ensure equitable practice, and to reduce the risk that interventions like screening programmes do not further exacerbate existing health inequalities.

Our study is a secondary data analysis aimed at estimating the proportion of individuals with SMI in Bangladesh, India, and Pakistan who were screened for NCDs and offered health risk modification advice. Furthermore, we explored socio-demographic factors associated with the likelihood of being screened for NCDs within this demographic.

## Methods

### Study design and setting

This is a secondary data analysis of a cross-sectional study which took place across three national mental health institutions—the National Institute of Mental Health and Hospital (NIMHH) in Dhaka, Bangladesh; the National Institute of Mental Health and Neurosciences (NIMHANS) in Bengaluru, India; the Institute of Psychiatry (IOP) Rawalpindi Medical University, Pakistan (Zavala et al., 2020, 2023).

### Participants

Participants aged ≥18 years who were diagnosed with SMI by their local doctor, and able to provide informed consent, were invited to participate in the study. SMI diagnoses were defined using the International Classification of Diseases 10th Revision (ICD-10), and this consisted of schizophrenia, schizotypal, and delusional disorders (F20–29), bipolar affective disorder (F30, F31), and severe depression with psychotic symptoms (F32·3, F33·3). The diagnosis was then confirmed using the Mini International Neuropsychiatric Interview (MINI) V6.0.

### Recruitment and consent

Recruitment took place between June 2019 and January 2022. Stratified random sampling was used to recruit a sample comprising 80% outpatients and 20% inpatients, as this has been reported as the usual proportion of outpatients and inpatients. At NIMHH and NIMHANS, potential participants were selected based on random number tables generated centrally. However, due to lower patient numbers at IOP, all patients attending IOP were approached for recruitment. Written and verbal information about the study was provided to all eligible participants. Those assessed as having capacity provided written consent and for those who had literacy difficulties, the statements were read aloud and a thumb impression was utilised instead of a signature.

### Data collection

Data were collected *via* face-to-face interviews with trained researchers using digital Qualtrics software. Each participant underwent a survey based on the WHO-STEPS survey (Bonita et al., 2003), which included questions about sociodemographic variables, screening and treatment for NCDs, health risk behaviours and health risk modification advice, followed by physical measurements such as blood pressure, height, weight, waist circumference and blood tests including HbA1c, non-fasting lipid profile and liver function tests.

### Dependent variables

Data were collected on the screening for NCDs and NCD risk factors through questions about whether individuals had undergone testing for diabetes, hypertension, and high cholesterol. Participants self-reported yes or no to this question to produce a dichotomous outcome.

The NCDs which we focussed on were diabetes and hypertension, we also included high cholesterol in the analysis because although this is not an NCD in itself, it is an important NCD risk factor. Hypertension was defined by blood pressure (BP) exceeding the cut-off (Systolic BP > 140 mmHg or Diastolic BP > 90 mmHg) when measured during the survey (measured three times), or by those who self-reported a diagnosis from a healthcare professional (Jordan et al., 2018).

Type 2 diabetes was defined by the $HbA_{1c}$ measurement ≥6.5% (48 mmol/mol) from the blood test carried out and those who self-reported (Herman and Fajans, 2010). Pre-diabetes, according to the American Diabetes Association (Basit et al., 2018), was defined by $HbA_{1c}$ between 5.7% and 6.4% (39–47 mmol/mol).

High cholesterol was defined as an LDL cholesterol concentration ≥100 mg/dL according to the blood test carried out during the survey and those who self-reported (WHO, 2015). All blood collection was carried out following the WHO STEPS surveillance manual (WHO, 2005).

Furthermore, health risk behaviour variables were based on participants' adherence to WHO guidance on particular behaviours. WHO guidance for physical activity stipulates that individuals should aim for 600 metabolic equivalents (MET)-min per week (Cleland et al., 2014), and with regard to diet, individuals are advised to consume at least five portions of fruit and vegetables per day (WHO, 2003, 2010). Also, a variable for smoking was measured using self-reported smoking status.

### Independent variables

To comprehensively assess the factors influencing NCD screening among individuals with severe mental illness (SMI), we explored several variables using the WHO-STEPS method: We included the BMI category because obesity and overweight are significant risk factors for NCDs. BMI was calculated using height and weight measurements, categorised using both international and Asian-specific WHO cut-off points (normal weight $18.5$ kg/m$^2$–$22.99$ kg/m$^2$; overweight $23$ kg/m$^2$–$24.99$ kg/m$^2$; obesity ≥$25$ kg/m$^2$) to account for the higher predisposition of South Asians to obesity-related NCDs. Height and weight were used to calculate BMI using the (weight (kg)/height (m)$^2$) formula, with categories defined using international WHO cut-off points (normal weight $18.5$ kg/m$^2$–$24.99$ kg/m$^2$; overweight $25$ kg/m$^2$–$29.99$ kg/m$^2$; obesity ≥$30$ kg/m$^2$) (World Health Organisation, 2000).Sex and age group were also considered, as these demographic factors can influence disease prevalence and health-seeking behaviours. Age groups were divided into four categories to capture variations across different life stages.

The type of SMI diagnosis was included, confirmed using the MINI International Neuropsychiatric Interview (MINI) V6.0. We also examined the duration of SMI, categorised into different time frames to understand the impact of illness duration on health screening.

Patient setting (inpatient *vs.* outpatient) was considered to see how the care environment influences health screening. The level of education was included to gauge the impact of health literacy on screening rates. Work status and income tertile were included as socioeconomic indicators, reflecting access to healthcare and financial ability to seek medical services. We also included the country to account for regional differences in healthcare infrastructure, policies, and cultural attitudes towards health, comparing data across Bangladesh, India, and Pakistan. Additional variables such as marital status, monthly household income, and the prevalence of other physical health conditions were also considered to provide a comprehensive analysis of factors associated with NCD screening.

### Statistical analysis

The initial study collected large amounts of data generating many research questions. A sample size calculation for one of the primary research questions, estimating the prevalence of diabetes, was presented in the research protocol (Zavala et al., 2020). A sample size calculation was not performed for the research questions covered by this secondary analysis.

All analyses were carried out using Stata Version 18.0. All analysis models were interpreted with statistical significance at the 5% level. Variables included in the models were identified *a priori* as likely to be assessed with screening and were individually tested for statistical significance, adjustment for multiple testing was not required. As the missingness of variables included within the model was minimal (<5%), as shown in Table 1, analysis models include complete cases only.

Continuous data were summarised using descriptive statistics (*n*, mean, standard deviation, median, interquartile range, minimum and maximum), while categorical data were reported as counts and percentages. These were summarised for each country and overall.

The rates of NCD and NCD risk factor screening, prevalence of NCDs and treatment of NCDs (diabetes, hypertension) and NCD risk factors (high cholesterol) within each country were described with corresponding 95% confidence intervals (see Table 2). We have used the term 'screening' to mean any identification or assessment of an NCD or NCD risk factor in the past reported by participants. This broader definition of screening provides an indication of screening practices in general rather than limiting the assessment to the presence of specific, dedicated NCD screening programmes. Prevalence of NCD was defined as either: a previous diagnosis of an NCD from a health professional (self-reported by the participant) or the participant being newly diagnosed from the SMI survey (this involved participants who had never been screened or did not have a previous diagnosis). Treated participants were defined as those who had been prescribed medication for the NCD, although a minority of participants may have received non-pharmacological treatment.

Health risk modification advice for diet, physical activity and smoking was recorded. The prevalence of receiving health modification advice for each country and overall was described with corresponding 95% confidence intervals. For each health risk, the prevalences of those who were meeting the WHO recommendations and those who received health modification advice were reported.

The associations between screening for NCDs and NCD risk factors and sociodemographic variables were investigated. A dichotomous (yes/no) variable was defined for participants who were screened for at least one of the NCDs mentioned previously. Logistic regression models were fitted, including the following sociodemographic variables: BMI category (WHO international cut-offs), sex, age group, SMI diagnosis, patient setting (inpatient or outpatient), level of education, work status, income tertile and country. The odds ratios (OR) were reported with corresponding 95% confidence intervals and *p*-values. Further models were fitted to observe interactions between the independent variables and the country. These models were assessed using a likelihood ratio test to compare to the model with no interaction terms.

## Results

### Participant characteristics

The majority of participants were male (59.1%), the average age was 35.8 years (SD = 11.9), and the most common age group was 25–39 years (Table 1). The most prevalent psychiatric diagnosis was schizophrenia and related disorders (44.7%). Pakistan had the greatest proportion of participants with higher/more than secondary level of education (41.8%). For both Bangladesh (45.0%) and Pakistan (43.3%), the lowest income tertile was the most common; however, in India, the middle-income tertile was most common (44.9%). The mean BMI was 25.1, and according to the WHO

**Table 1.** Participant characteristics

| | Bangladesh (*n* = 1,500) | India (*n* = 1,175) | Pakistan (*n* = 1,314) | Overall (*n* = 3,989) |
|---|---|---|---|---|
| **Sex, *n* (%)** | | | | |
| *Number with data* | *1,500 (100)* | *1,175 (100)* | *1,314 (100)* | *3,989 (100)* |
| Male | 915 (61.0) | 648 (55.1) | 796 (60.6) | 2,359 (59.1) |
| Female | 585 (39.0) | 527 (44.9) | 518 (39.4) | 1,630 (40.9) |
| **Age (years)** | | | | |
| *n* (%) | 1,500 (100) | 1,175 (100) | 1,314 (100) | 3,989 (100) |
| Mean (SD) | 31.5 (10.8) | 38.8 (11.2) | 38.1 (12.3) | 35.8 (11.9) |
| Median (IQR) | 30.0 (23.0–38.0) | 38.0 (30.0–46.0) | 36.0 (28.0–45.0) | 35.0 (26.0–44.0) |
| Min, Max | 18.0, 76.0 | 18.0, 81.0 | 18.0, 84.0 | 18.0, 84.0 |
| **Age group, *n* (%)** | | | | |
| *Number with data* | *1,500 (100)* | *1,175 (100)* | *1,314 (100)* | *3,989 (100)* |
| 18–24 years | 434 (28.9) | 123 (10.5) | 159 (12.1) | 716 (17.9) |
| 25–39 years | 732 (48.8) | 538 (45.8) | 603 (45.9) | 1873 (47.0) |
| 40–54 years | 263 (17.5) | 386 (32.9) | 402 (30.6) | 1,051 (26.3) |
| 55+ years | 71 (4.7) | 128 (10.9) | 150 (11.4) | 349 (8.7) |
| **SMI, *n* (%)** | | | | |
| *Number with data* | *1,500 (100)* | *1,175 (100)* | *1,314 (100)* | *3,989 (100)* |
| Schizophrenia and related disorders | 935 (62.3) | 673 (57.3) | 176 (13.4) | 1784 (44.7) |
| Major depressive disorder with psychotic features | 77 (5.1) | 63 (5.4) | 601 (45.7) | 741 (18.6) |
| Bipolar disorder | 488 (32.5) | 439 (37.4) | 537 (40.9) | 1,464 (36.7) |
| **Duration of the SMI, *n* (%)** | | | | |
| *Number with data* | *1,500 (100)* | *1,175 (100)* | *1,314 (100)* | *3,989 (100)* |
| ≤2 years | 436 (29.1) | 215 (18.3) | 289 (22.0) | 940 (23.6) |
| 3–5 years | 457 (30.5) | 266 (22.6) | 320 (24.4) | 1,043 (26.1) |
| 6–10 years | 332 (22.1) | 299 (25.4) | 299 (22.8) | 930 (23.3) |
| >10 years | 271 (18.1) | 359 (30.6) | 399 (30.4) | 1,029 (25.8) |
| Do not know/do not remember | 4 (0.3) | 36 (3.1) | 7 (0.5) | 47 (1.2) |
| **Setting, *n* (%)** | | | | |
| *Number with data* | *1,500 (100)* | *1,175 (100)* | *1,314 (100)* | *3,989 (100)* |
| Inpatient | 313 (20.9) | 264 (22.5) | 122 (9.3) | 699 (17.5) |
| Outpatient | 1,187 (79.1) | 911 (77.5) | 1,192 (90.7) | 3,290 (82.5) |
| **Highest level of education, *n* (%)** | | | | |
| *Number with data* | *1,500 (100)* | *1,174 (99.9)* | *1,312 (99.8)* | *3,986 (99.9)* |
| No formal education | 151 (10.1) | 141 (12.0) | 257 (19.6) | 549 (13.8) |
| Primary education | 842 (56.1) | 401 (34.2) | 234 (17.8) | 1,477 (37.1) |
| Secondary education | 234 (15.6) | 228 (19.4) | 273 (20.8) | 735 (18.4) |
| Higher/more than secondary | 273 (18.2) | 404 (34.4) | 548 (41.8) | 1,225 (30.7) |
| **Work status (past 12 months), *n* (%)** | | | | |
| *Number with data* | *1,500 (100)* | *1,174 (99.9)* | *1,309 (99.6)* | *3,983 (99.8)* |
| Employed | 439 (29.3) | 522 (44.5) | 507 (38.7) | 1,468 (36.9) |
| Unemployed | 595 (39.7) | 227 (19.3) | 291 (22.2) | 1,113 (27.9) |
| Other[1] | 466 (31.1) | 425 (36.2) | 511 (39.0) | 1,402 (35.2) |

(*Continued*)

**Table 1.** (*Continued*)

| | Bangladesh (*n* = 1,500) | India (*n* = 1,175) | Pakistan (*n* = 1,314) | Overall (*n* = 3,989) |
|---|---|---|---|---|
| **Monthly household income (USD)** | | | | |
| *n* (%) | 1,497 (99.8) | 1,032 (87.8) | 1,307 (99.5) | 3,836 (96.2) |
| Mean (SD) | 224.6 (352.1) | 305.2 (861.4) | 198.6 (199.8) | 237.4 (513.1) |
| Median (IQR) | 176.7 (117.8–235.5) | 158.9 (66.2–264.8) | 148.6 (89.2–237.8) | 158.9 (105.9–264.8) |
| Min, Max | 11.8, 11,777.2 | 0.0, 16,551.9 | 0.0, 2,972.7 | 0.0, 16,551.9 |
| **Income tertile, *n* (%)** | | | | |
| *Number with data* | *1,497 (99.8)* | *1,032 (87.8)* | *1,307 (99.5)* | *3,836 (96.2)* |
| Low | 674 (45.0) | 349 (33.8) | 566 (43.3) | 1,589 (41.4) |
| Middle | 485 (32.4) | 463 (44.9) | 315 (24.1) | 1,263 (32.9) |
| High | 338 (22.6) | 220 (21.3) | 426 (32.6) | 984 (25.7) |
| **Marital status, *n* (%)** | | | | |
| *Number with data* | *1,500 (100)* | *1,175 (100)* | *1,314 (100)* | *3,989 (100)* |
| Never married | 539 (35.9) | 349 (29.7) | 417 (31.7) | 1,305 (32.7) |
| Currently married | 818 (54.5) | 711 (60.5) | 747 (56.8) | 2,276 (57.1) |
| Ever married[2] | 143 (9.5) | 115 (9.8) | 150 (11.4) | 408 (10.2) |
| **BMI** | | | | |
| *n* (%) | *1,497 (99.8)* | *1,161 (98.8)* | *1,304 (99.2)* | *3,962 (99.3)* |
| Mean (SD) | *24.0 (4.5)* | *25.6 (5.3)* | *25.9 (6.1)* | *25.1 (5.4)* |
| Median (IQR) | *23.6 (21.1–26.5)* | *25.0 (22.1–28.5)* | *25.3 (21.8–29.3)* | *24.5 (21.6–28.1)* |
| Min, Max | *14.0, 83.8* | *13.0, 96.9* | *9.7, 65.5* | *9.7, 96.9* |
| **BMI (WHO international classification), *n* (%)** | | | | |
| *Number with data* | *1,497 (99.8)* | *1,161 (98.8)* | *1,304 (99.2)* | *3,962 (99.3)* |
| Underweight (BMI < 18.5) | 129 (8.6) | 66 (5.7) | 113 (8.7) | 308 (7.8) |
| Normal weight (BMI 18.5–24.9) | 817 (54.6) | 506 (43.6) | 501 (38.4) | 1824 (46.0) |
| Overweight (BMI 25.0–29.9) | 405 (27.1) | 384 (33.1) | 406 (31.1) | 1,195 (30.2) |
| Obesity (BMI ≥ 30.0) | 146 (9.8) | 205 (17.7) | 284 (21.8) | 635 (16.0) |
| **BMI (WHO Asian cut-offs), *n* (%)** | | | | |
| *Number with data* | *1,497 (99.8)* | *1,161 (98.8)* | *1,304 (99.2)* | *3,962 (99.3)* |
| Underweight (BMI < 18.5) | 129 (8.6) | 66 (5.7) | 113 (8.7) | 308 (7.8) |
| Normal weight (BMI 18.5–22.9) | 523 (34.9) | 290 (25.0) | 327 (25.1) | 1,140 (28.8) |
| Overweight (BMI 23.0–24.9) | 294 (19.6) | 216 (18.6) | 174 (13.3) | 684 (17.3) |
| Obesity (BMI ≥ 25.0) | 551 (36.8) | 589 (50.7) | 690 (52.9) | 1830 (46.2) |
| **Waist circumference (cm)** | | | | |
| **Males** | | | | |
| *n* (% of males) | 915 (100) | 640 (98.8) | 786 (98.7) | 2,341 (99.2) |
| Mean (SD) | 83.3 (11.2) | 90.3 (12.3) | 89.9 (16.8) | 87.4 (14.0) |
| Median (IQR) | 84.5 (75.8–90.5) | 90.0 (82.0–98.0) | 89.2 (81.0–99.0) | 87.5 (79.2–95.0) |
| Min, Max | 35.5, 163.3 | 50.6, 140.0 | 30.3, 191.2 | 30.3, 191.2 |
| **Females** | | | | |
| *n* (% of females) | 582 (99.5) | 516 (97.9) | 507 (97.9) | 1,605 (98.5) |
| Mean (SD) | 84.3 (12.2) | 87.2 (13.2) | 90.6 (22.2) | 87.2 (16.5) |
| Median (IQR) | 85.5 (76.5–91.5) | 87.5 (79.0–95.3) | 92.0 (78.2–105.0) | 87.6 (78.0–96.5) |
| Min, Max | 46.5, 174.5 | 40.0, 130.2 | 30.0, 193.2 | 30.0, 193.2 |

(*Continued*)

**Table 1.** (*Continued*)

|  | Bangladesh (*n* = 1,500) | India (*n* = 1,175) | Pakistan (*n* = 1,314) | Overall (*n* = 3,989) |
|---|---|---|---|---|
| **Abdominal obesity, *n* (%)** | | | | |
| **Males** | | | | |
| *Number with data* | *915 (100)* | *640 (98.8)* | *786 (98.7)* | *2,341 (99.2)* |
| No abdominal obesity | 666 (72.8) | 300 (46.9) | 398 (50.4) | 1,364 (58.3) |
| Has abdominal obesity | 249 (27.2) | 340 (53.1) | 388 (49.4) | 977 (41.7) |
| **Females** | | | | |
| *Number with data* | *582 (99.5)* | *516 (97.9)* | *507 (97.9)* | *1,605 (98.5)* |
| No abdominal obesity | 189 (32.5) | 138 (26.7) | 133 (26.2) | 460 (28.7) |
| Has abdominal obesity | 393 (67.5) | 378 (73.3) | 374 (73.8) | 1,145 (71.3) |
| **Males and Females** | | | | |
| *Number with data* | *1,497 (99.8)* | *1,156 (98.4)* | *1,293 (98.4)* | *3,946 (98.9)* |
| No abdominal obesity | 855 (57.1) | 438 (37.9) | 531 (41.1) | 1824 (46.2) |
| Has abdominal obesity | 642 (42.9) | 718 (62.1) | 762 (58.9) | 2,122 (53.8) |

[1]Other includes: homemaker, student, and retired.
[2]Ever married includes: widowed, separated, and divorced.
SD, Standard deviation. IQR, Interquartile range. Min, minimum. Max, maximum.

international cut-offs, the most common BMI category was normal weight (46.0%), which is contrasted by the Asian-specific cut-off values, which show that obesity was the most common category (46.2%). The majority of male and female patients were found to have abdominal obesity (53.8%) as per Asian cutoffs.

### Screening and diagnosis of NCDs and NCD risk factors according to country

Screening rates (proportion of people who have been screened in the past) were variable across the three countries, and also varied for each NCD and NCD risk factor (Table 2). For diabetes and high cholesterol, screening rates were highest in India (30.7% [28.1–33.4] and 14.8% [12.9–17.0], respectively), and hypertension screening was most common in Pakistan (60.9% [58.2–63.5]). High cholesterol was consistently least screened across all three countries (Bangladesh = 4.1% [3.2–5.2], India = 14.8% [12.9–17.0], Pakistan = 9.6% [8.1–11.3]), and hypertension was the most screened NCD (Bangladesh = 52.5% [50.0–55.1], India = 43.1% [40.3–45.9], Pakistan = 60.9% [58.2–63.5]). High cholesterol was the most prevalent self-reported NCD risk factor in Bangladesh (23.0% [14.1–35.2]) and Pakistan (42.1% [33.7–50.9]), and diabetes was the most prevalent self-reported NCD in India (26.0% [21.8–30.8]). High cholesterol was the most prevalent NCD risk factor that was newly diagnosed through testing in all three countries (Bangladesh = 45.6% [43.0–48.2]), India = 45.5% [42.4–48.7], Pakistan = 54.2% [51.4–57.0]). Diabetes was the most common NCD for which patients received treatment in all three countries (Bangladesh = 71.4% [56.0–83.1], India = 79.8% [70.4–86.8], Pakistan = 64.5% [53.1–74.4]).

### Prevalence of health risk modification advice given

Only 4.9% (4.3–5.6) of participants from across the three countries adhered to WHO recommendations for fruit and vegetable consumption (Table 3). However, dietary advice was the most common health risk modification advice provided across all three countries (34.2% [32.4–36.1]). India was the country that provided the most

health advice for all three health risk behaviours (diet = 66.7% [62.1–71.1], physical activity = 71.5% [67.0–75.6], smoking = 17.1% [13.8–21.0]). Conversely, Bangladesh provided the least health advice across all three health risk behaviours (diet = 17.8% [15.8–20.0], physical activity = 12.0% [10.3–13.8], smoking = 9.8% [8.3–11.5]).

### Association of sociodemographic variables with screening of NCDs and NCD risk factors

From the 3,989 recruited participants, 3,808 were included in the binomial logistic regression model displayed in Table 4. The specific reasons for exclusion are outlined in the flowchart which can be found in the Supplementary Appendix.

Participants with BMI less than 18.5 kg/m$^2$ (underweight) were less likely to be screened for at least one NCD or NCD risk factor (OR = 0.64, 95% CI:0.49 to 0.83) as compared to those with normal weight (Table 4), and as BMI increased, the participants were more likely to be screened. For example, participants with obesity were 53% more likely to be screened for an NCD or NCD risk factor compared to those with normal weight (OR = 1.53, 95% CI:1.25 to 1.89). Female participants were 29% more likely to be screened for NCDs or NCD risk factors compared to their male counterparts (OR = 1.29, 95% CI: 1.06 to 1.58). Participants older than 55 years were three times more likely to be screened for NCDs or NCD risk factors compared to those between 18 and 24 years (OR = 3.74, 95% CI:2.75 to 5.08). Participants with bipolar disorder were more likely to be screened for NCDs or NCD risk factors compared to those diagnosed with schizophrenia-type disorder (OR = 0.84, 95% CI: 0.71 to 0.98).

Participants with higher or more than a secondary level of education were 61% more likely to be screened for NCDs or NCD risk factors compared to those with no formal education (OR = 1.61, 95% CI: 1.27 to 2.04). Unemployed participants were 20% less likely to be screened for NCDs or NCD risk factors compared to those with a job (OR = 0.80, 95% CI:0.67 to 0.95). Also, participants in the highest income tertile were more likely to be screened for NCDs or NCD risk factors compared to participants in the lowest tertile (OR = 1.21, 95% CI:1.01 to 1.44). Indian

**Table 2.** Screening, prevalence and treatment of NCDs and NCD risk factors according to country

| | Screening | Prevalence | | Treatment |
|---|---|---|---|---|
| | Proportion of people who have been screened in the past % (95% CI) | Proportion of people self-reported % (95% CI) | Proportion of people newly diagnosed through testing[1]% (95% CI) | Proportion of people who have received treatment[2]% (95% CI) |
| **Bangladesh** | | | | |
| Diabetes | 334/1500 | 42/334 | 85/1391 | 30/42 |
| | 22.3 (20.2–24.4) | 12.6 (9.4–16.6) | 6.1 (5.0–7.5) | 71.4 (56.0–83.1) |
| Hypertension | 788/1500 | 89/788 | 54/1411 | 45/89 |
| | 52.5 (50.0–55.1) | 11.3 (9.3–13.7) | 3.8 (2.9–5.0) | 50.6 (40.3–60.8) |
| High cholesterol | 61/1500 | 14/61 | 645/1415 | 5/14 |
| | 4.1 (3.2–5.2) | 23.0 (14.1–35.2) | 45.6 (43.0–48.2) | 35.7 (15.5–62.7) |
| **India** | | | | |
| Diabetes | 361/1175 | 94/361 | 70/871 | 75/94 |
| | 30.7 (28.1–33.4) | 26.0 (21.8–30.8) | 8.0 (6.4–10.0) | 79.8 (70.4–86.8) |
| Hypertension | 506/1175 | 85/506 | 57/1082 | 47/85 |
| | 43.1 (40.3–45.9) | 16.8 (13.8–20.3) | 5.3 (4.1–6.8) | 55.3 (44.6–65.5) |
| High cholesterol | 174/1175 | 37/174 | 433/951 | 14/37 |
| | 14.8 (12.9–17.0) | 21.3 (15.8–28.0) | 45.5 (42.4–48.7) | 37.8 (23.7–54.4) |
| **Pakistan** | | | | |
| Diabetes | 372/1314 | 76/372 | 45/1201 | 49/76 |
| | 28.3 (25.9–30.8) | 20.4 (16.6–24.8) | 3.7 (2.8–5.0) | 64.5 (53.1–74.4) |
| Hypertension | 800/1314 | 293/800 | 38/1014 | 113/293 |
| | 60.9 (58.2–63.5) | 36.6 (33.4–40.0) | 3.7 (2.7–5.1) | 38.6 (33.1–44.3) |
| High cholesterol | 126/1314 | 53/126 | 662/1222 | 17/53 |
| | 9.6 (8.1–11.3) | 42.1 (33.7–50.9) | 54.2 (51.4–57.0) | 32.1 (20.9–45.8) |

[1]Denominator includes participants who had the blood test that have never been screened for an NCD or did not self-report an NCD.
[2]Denominator is the number of people who have self-reported to have a diagnosis of the condition.

**Table 3.** Prevalence of health behaviours and corresponding health risk modification advice given

| Countries | All | Bangladesh | India | Pakistan |
|---|---|---|---|---|
| **Diet** | | | | |
| Complying with WHO recommendations for fruit/veg % (95% CI) | 196/3989 | 117/1500 | 30/1175 | 49/1314 |
| | 4.9 (4.3–5.6) | 7.8 (6.5–9.3) | 2.6 (1.8–3.6) | 3.7 (2.8–4.9) |
| Has received advice about altering diet in the past?[1]% (95% CI) | 836/2443 | 231/1297 | 281/421 | 324/725 |
| | 34.2 (32.4–36.1) | 17.8 (15.8–20.0) | 66.7 (62.1–71.1) | 44.7 (41.1–48.3) |
| **Physical activity** | | | | |
| Complying with WHO recommendations for physical activity % (95% CI) | 1817/3989 | 889/1500 | 388/1175 | 540/1314 |
| | 45.6 (44.0–47.1) | 59.3 (56.8–61.7) | 33.0 (30.4–35.8) | 41.1 (38.5–43.8) |
| Has received advice about physical activity in the past?[1]% (95% CI) | 655/2443 | 155/1297 | 301/421 | 199/725 |
| | 26.8 (25.1–28.6) | 12.0 (10.3–13.8) | 71.5 (67.0–75.6) | 27.4 (24.3–30.8) |
| **Smoking** | | | | |
| Prevalence of current smokers % (95% CI) | 1394/3969 | 585/1499 | 283/1167 | 526/1303 |
| | 35.1 (33.7–36.6) | 39.0 (36.6–41.5) | 24.3 (21.9–26.8) | 40.4 (37.7–43.1) |
| Has received advice about smoking in the past?[1]% (95% CI) | 301/2443 | 127/1297 | 72/421 | 102/725 |
| | 12.3 (11.1–13.7) | 9.8 (8.3–11.5) | 17.1 (13.8–21.0) | 14.1 (11.7–16.8) |

[1]Due to the questionnaire format, if a participant had not visited a health professional in the past 12 months, they were not asked to answer the follow-up questions regarding health modification advice (*n* = 1,546).
CI, confidence interval.

**Table 4.** Association of sociodemographic variables with screening of NCDs and NCD risk factors

| | Self-reported screening of at least one NCD or NCD risk factor | OR (95% CI) | *p* Value |
|---|---|---|---|
| **BMI category** | | | |
| Normal weight | 923/1760 (52.4) | Reference | |
| Underweight | 125/298 (41.9) | 0.64 (0.49–0.83) | *p* < 0.001 |
| Overweight | 695/1149 (60.5) | 1.22 (1.04–1.43) | *p* = 0.013 |
| Obesity | 406/601 (67.6) | 1.53 (1.25–1.89) | *p* < 0.001 |
| **Sex** | | | |
| Male | 1225/2283 (53.7) | Reference | |
| Female | 924/1525 (60.6) | 1.29 (1.06–1.58) | *p* = 0.013 |
| **Age group** | | | |
| 18–24 years | 307/695 (44.2) | Reference | |
| 25–39 years | 969/1797 (53.9) | 1.48 (1.23–1.79) | *p* < 0.001 |
| 40–54 years | 636/988 (64.4) | 2.38 (1.91–2.96) | *p* < 0.001 |
| 55+ years | 237/328 (72.3) | 3.74 (2.75–5.08) | *p* < 0.001 |
| **SMI diagnosis** | | | |
| Bipolar disorder | 827/1393 (59.4) | Reference | |
| Major depressive disorder with psychotic features | 455/720 (63.2) | 0.94 (0.76–1.16) | *p* = 0.575 |
| Schizophrenia-type disorder | 867/1695 (51.2) | 0.84 (0.71–0.98) | *p* = 0.026 |
| **Setting** | | | |
| Inpatient | 350/662 (52.9) | Reference | |
| Outpatient | 1799/3146 (57.2) | 0.99 (0.82–1.18) | *p* = 0.889 |
| **Level of education** | | | |
| No formal education | 308/518 (59.5) | Reference | |
| Primary education | 725/1422 (51.0) | 0.99 (0.79–1.24) | *p* = 0.912 |
| Secondary education | 392/699 (56.1) | 1.23 (0.96–1.59) | *p* = 0.100 |
| Higher/more than secondary | 724/1169 (61.9) | 1.61 (1.27–2.04) | *p* < 0.001 |
| **Work status (past 12 months)** | | | |
| Employed | 806/1424 (56.6) | Reference | |
| Unemployed | 534/1064 (50.2) | 0.80 (0.67–0.95) | *p* = 0.013 |
| Other[1] | 809/1320 (61.3) | 0.97 (0.77–1.21) | *p* = 0.761 |
| **Income tertile** | | | |
| Low | 853/1573 (54.2) | Reference | |
| Middle | 678/1254 (54.1) | 1.03 (0.88–1.21) | *p* = 0.684 |
| High | 618/981 (63.0) | 1.21 (1.01–1.44) | *p* = 0.037 |
| **Country** | | | |
| Bangladesh | 822/1494 (55.0) | Reference | |
| India | 501/1023 (49.0) | 0.51 (0.42–0.61) | *p* < 0.001 |
| Pakistan | 826/1291 (64.0) | 0.91 (0.75–1.11) | *p* = 0.361 |

[1]Other includes: homemaker, student, and retired.
OR, odds ratio. CI, confidence interval.

participants were 49% less likely to be screened for NCDs or NCD risk factors compared to participants from Bangladesh (OR = 0.51, 95% CI:0.42 to 0.61).

Additional individual models were fitted, including an interaction term between country and each variable. When comparing to a model with no interactions, the likelihood ratio test identified the variables sex (*p* = 0.003), patient setting (*p* = 0.05), SMI diagnosis (*p* < 0.001), work status (*p* = 0.001) and income (*p* = 0.023) as having a significant improvement to the fit of the model when an interaction effect with country is included. The direction and effect of these interaction terms can be found in the Supplementary Appendix.

## Discussion

### Summary of the findings

Our results reveal significant gaps in NCD and NCD risk factor screening practices and health risk modification advice provision across different demographic and clinical subgroups. High cholesterol was underdiagnosed across all three national institution samples, and consequently, the most common newly diagnosed condition in all three samples. Being female, having a higher BMI, increased age and higher educational attainment were positively associated with screening and the provision of health risk modification advice was most common in the Indian sample.

### Explanation of the findings

A notable finding in our study is the considerable variability in screening rates for different NCDs across the three national institution samples examined. While hypertension screening appears relatively robust, hypercholesterolemia had low screening rates across all three samples. This inconsistency underscores the urgent need for standardised screening protocols and strengthened healthcare infrastructure to ensure comprehensive NCD surveillance and early detection, especially among people with SMI.

Furthermore, our analysis identifies several socio-demographic and clinical factors associated with NCD screening disparities. Higher education levels, employment status, and income were positively correlated with screening likelihood, highlighting the role of socio-economic determinants in healthcare access and utilisation. This is mirrored by Legesse et al., 2022 who identified that people with secondary education were more than five times more likely to be screened for NCDs. This is likely due to higher educational attainment being linked to higher health literacy (Jansen et al., 2018). Studies have shown that increased knowledge of risk factors of NCDs is the most important determinant in the utilisation of health screening (Legesse et al., 2022). Multiple reviews also identified that the financial cost of attending screening programmes was a key barrier to utilisation of such health services, which would therefore impact those of lower socioeconomic status more than those with higher income (Islam et al., 2017; Lim and Ojo, 2017). People with lower income often have more pressing healthcare issues that require their limited financial resources. Therefore, due to the out-of-pocket payments that attending screening will require, this is often deprioritised (Lim and Ojo, 2017).

Conversely, certain clinical factors such as psychiatric diagnosis also influenced screening rates, underscoring the importance of integrated care models that address both physical and mental health needs holistically. People with bipolar disorders were more likely to be screened than patients with schizophrenia-type disorders, which may be explained by the increased likelihood of having more physical health multimorbidity, as well as the reduced prevalence of chronic, undertreated negative and cognitive symptoms that present additional barriers to attending screening (Launders et al., 2022). Patient setting did not influence screening rates which may indicate that inpatients do not have comparatively greater access to NCD screening compared to people in the community. Additionally, inpatients below the poverty line at NIMHANS were provided with an individualised diet after a dietetic referral, and all inpatients at NIMH and IOP received regular meals however these were standardised and not tailored. The survey did not collect further information about hospital food provision as the study was mainly focused around diet in general however, it may be pertinent to explore the provision of hospital meals further to investigate if they comply with WHO standards.

### Comparison with other populations

This cross-sectional survey focused on people with SMI therefore direct comparisons to the general population cannot be made, however indirect comparisons to the WHO Stepwise survey findings illustrate the potential disparities between people with SMI and the general population.

The prevalence of diabetes screening in our study ranged from 22% to 30%, which is lower than the screening prevalence in people with SMI in the UK (65%) (Mitchell and Hardy, 2013) and similar to the screening prevalence seen in the general population in Bangladesh (25.1%)(Islam et al., 2018), India (26.3%) (ICMR-NCDIR, 2020), and Pakistan (21.5%) (PHRC, 2016). Hypertension screening in our study was lower (43–60%) compared to people with SMI in the UK (84%) (Mitchell and Hardy, 2013). It was lower than screening in the general population in Bangladesh (70.1%) (Islam et al., 2018), but comparable to India (52.0%) (ICMR-NCDIR., 2020), and Pakistan (54.7%) (PHRC, 2016). Regarding cholesterol screening, the prevalence in our study (4% to 15%) is also lower than the SMI population in the UK (72%) (Mitchell and Hardy, 2013) but is similar to the general population in Bangladesh (4.6%) (Islam et al., 2018), India (6.4%) (ICMR-NCDIR., 2020), and Pakistan (6.2%)(PHRC, 2016). It should be noted that nearly half of the participants (47.0%) in the survey were aged between 25 and 39 years old which is a considerably larger proportion than some of the general population surveys (for example in Bangladesh this age group made up 39.7% of participants), therefore this may explain some of the lower screening rates seen in our survey.

The findings above highlight significant disparities in the screening rates for diabetes, hypertension, and lipid disorders among people with SMI accessing these LMIC tertiary institutions compared to HICs, as well as between people with SMI and the general population (Sud, 2021). These disparities can be attributed to several factors. In LMICs, there are often limited healthcare resources, insufficient training for healthcare providers, and lower healthcare funding, which collectively results in reduced access to regular health screening and lack of structured programmes (Martinez et al., 2024). Additionally, cultural stigma surrounding mental illness in many LMICs, especially in South Asia, can further impede individuals with SMI from seeking necessary healthcare support for both mental illness and comorbid physical health problems, therefore creating an additional barrier to managing NCDs (Vaishnav et al., 2023). In contrast, HICs generally have more robust healthcare infrastructures, better funding, and structured screening programs, which contribute to higher screening rates for NCDs among both the general population and individuals with SMI.

### Implications

In resource-stricken healthcare settings such as those of LMICs, prevention of NCDs has been found to be cost effective according to a scoping review of studies in South East Asia (Nguyen et al., 2023). For example, prevention of cardiovascular disease by treating hypertension was found to be the most cost-effective intervention, and for diabetes, all types of screening were found to be cost-effective (Nguyen et al., 2023). However, further research assessing

the cost-effectiveness of such interventions in people with SMI is needed.

### Strengths and limitations

Several limitations to this study should be acknowledged. As a cross-sectional study, it cannot establish causality, meaning we cannot determine whether certain exposures cause increased NCD screening or *vice versa*. Also, there was no control group (people without SMI), direct comparisons of screening rates with the general population cannot be made. While the survey and analysis cover important factors associated with NCD risks, we were not able to address barriers such as stigma, which may impact NCD screening. There is a risk of residual confounding, where unmeasured variables may influence the observed relationships. However, this is inherent to all cross-sectional studies and not unique to this research. In addition, a large proportion of the data about screening, prevalence and treatment of NCDs was based upon individual participants' own recollection; therefore, this is prone to recall bias. Also, this study focussed specifically on two NCDs (hypertension and diabetes) and one NCD risk factor (high cholesterol) as they were considered to be the three major components of metabolic syndrome, which is seen extensively in this population. Another limitation of this study is that our analysis was not based on a population-level survey (*e.g.* household survey). The survey was conducted among individuals who attended tertiary institutions and provided complete data, which is not representative of the entire population of people with SMI. Although the sample was predominantly an outpatient population (80%), it did not include individuals accessing other mental health services, such as community clinics. However, in South Asia, such clinics are still rare and a large proportion of people will attend outpatient services at a tertiary mental health institution directly as the "first point of care" (Khemani et al., 2020; Andary et al., 2023). Additionally, many people with SMI may not have access to tertiary institutions, which could potentially lead to an overestimation of screening rates. This suggests that the true screening estimates for the broader population of people with SMI in these countries might actually be lower than reported in this study.

The study has several notable strengths. It is one of the few detailed investigations into the screening of NCDs and NCD risk factors among people with SMI in South Asia, an often under-represented population in medical research. With 3,989 participants, the large sample size enables robust analysis of associations between sociodemographic variables and the likelihood of being screened. This comprehensive study provides invaluable data on the comparative prevalence of physical health screening and interventions in the South Asian SMI population, offering a critical scaffold for making comparisons with other global populations and developing strategies to improve healthcare outcomes in this vulnerable group.

### Conclusion

This study indicates there are low levels of screening for NCDs in people with SMI accessing tertiary institutions in South Asia, despite the evidence identifying this population as most at risk of developing these conditions. Males with SMI, people of the lowest income tertile and lowest level of educational attainment were least likely to have been screened. By identifying some of these key sub-groups, policymakers and healthcare systems can utilise this

information for designing and implementing targeted screening programmes, as well as the provision of tailored health risk modification advice. Improving health literacy is also key to ensuring such interventions lead to lasting impacts and ultimately reducing the mortality gap between people with SMI and the general population.

**Open peer review.** To view the open peer review materials for this article, please visit http://doi.org/10.1017/gmh.2026.10157.

**Supplementary material.** The supplementary material for this article can be found at http://doi.org/10.1017/gmh.2026.10157.

**Data availability statement.** The data that support the findings of this study are available on request from the corresponding author, K Appuhamy. The data are not publicly available due to information that could compromise the privacy of participants involved.

**Acknowledgements.** The authors would like to acknowledge Abu Musa Robin, Anjuman Jum Tithi, Tanvir Arafat, Lipon Saha, Tasmia Rahman, Dr. Khaleda Islam, Dr. Mamun, Dr.Bappi, from the team in Bangladesh; Archith Krishna, Sathish Kumar, Neeta P.S, Venkatalakshmi, Bhuvneshwari L, Manjunatha S, Sobin George, Krishna Jayanthi from the team in India; and Rubab Ayesha, Nida Afsheen, Najma Hayat, Zaheen Amin, Aniqa Maryam from the team in Pakistan for conducting all the interviews and physical measurements, providing technical and managerial support. The authors also would like to thank all the participants who consented and provided their time to complete the interview.

**Author contribution.** NS, KS JB, PM, AN and AT conceived the primary study from which this secondary data collection is based and NS, JB, KS and AT were involved in funding acquisition. KA, FW, AM, GZ conceptualised and designed this study. HA, FA, RH, KPM were involved in the primary data collection. JB and GZ carried out data curation for archiving and access. FW, AM, RH, KPM were also involved in data curation and formal analysis. BP and VS provided additional editing and review support, MA and DS provided patient and primary care perspectives during the review process. KA, FW, AM, GZ, RIH, OG contributed to the original draft and all authors revised and approved the final manuscript.

**Financial support.** This research was funded by the National Institute for Health Research (NIHR) (Grant: GHRG 17/63/130:) using UK aid from the UK Government to support global health research.
The views expressed in this publication are those of the author(s) and not necessarily those of the NIHR or the UK Department of Health and Social Care.

**Competing interests.** DS is an expert advisor to the NICE Centre for Guidelines; the views expressed are the authors' and not those of NICE.

**Ethical standards disclosure.** This study was conducted according to the guidelines laid down in the Declaration of Helsinki and all procedures involving research study participants were approved by the ethics committees of the Department of Health Sciences, University of York, UK (HSRGC-3/17); the Centre for Injury Prevention and Research, Bangladesh (CIPRB/ERC/2OI 8/003); the Institute Ethics Committee, National Institute of Mental Health and Neurosciences, India (BEH.SC.DIV 20/19); the Health Ministry Screening Committee, India (HMSC12/18); and the National Bioethics Committee, Pakistan (4-18/NBC-413/19). Written informed consent was obtained from all patients.

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
