## [Reviewer Report]

Title

The definition: “Severe Mental Disorders” may be preferred to “Severe Mental Illness” (as per World Mental Health Report).

Abstract

Hypercholesterolemia may not classify as an NCDs per se, like diabetes.

Introduction

Factors leading to vulnerability to NCDs of people with severe mental disorders rightly include poor diet, physical inactivity, obesity, genetic and environmental factors, healthcare inequalities, as well as adverse effects from antipsychotic treatment. No reference is made here to stigma as a barrier to accessing healthcare an a equal basis with those without severe mental disorders.

Methods

Participants were all recruited in national mental health institutions. No data is presented on how people with severe mental disorders may access treatment for NCDs in other health devices, such as general hospitals, PHC, community clinics, etc. It would be more interesting to compare access to treatment for NCDs in institutional settings vs community settings.

Limitations

Poor diet is identified as a factor leading to additional vulnerability vis-a -vis NCDs. Are recruited participants (inpatients) provided with meals by the three hospitals? Are meals individualized to their needs? (salt, fats, fibers, etc).

---

## [Reviewer Report]

Design vs claims: This is a secondary analysis of an institutional sample, not a population-level epidemiological study, yet the conclusions read as if the study was designed to measure national screening coverage.

WHO-STEPS scope: WHO-STEPS is a tool for ‘risk factor surveillance’ , not for diagnosing NCDs or evaluating screening programs, using it here is reasonable — but turning it into a proxy for “screening adequacy” may be inappropriate.

What was actually measured: The study asked participants if they had “ever been tested.” That reflects the ‘prevalence of prior testing’, which may include opportunistic, diagnostic, or incidental checks. This is not the same as structured, preventive ‘screening’ for its own sake — though the paper sometimes uses the terms interchangeably.

Representativeness: The sample (patients attending three national psychiatric institutions) is large and valuable, but it cannot represent all people with SMI in South Asia. This should be mentioned.

Framing: The descriptive findings are strong enough on their own and can merit a paper. The repeated claim of “inadequate screening” may well be true but this paper does not answer that.

---

## [Editor Report]

Dear authors,

Thank you for submitting your manuscript examining screening for NCDs among people with severe mental disorders. The reviewers agree that the topic is of substantial public health importance and that the data collected from psychiatric institutions in South Asia represent a valuable contribution to the literature. However, several concerns need to be addressed before the paper can be considered for publication. Reviewers highlighted issues with conceptual framing (e.g., appropriate terminology and classification of NCDs), methodological clarity (e.g., institutional versus community recruitment, representativeness), and interpretative overreach (e.g., conflating “screening adequacy” with “history of testing”). They also suggested integrating key contextual factors—such as stigma and dietary environments in institutional settings—that influence access to care. You can find reviewers' comments below.